# Some Results for Split Equality Equilibrium Problems in Banach Spaces

**Zhaoli Ma [1], Lin Wang [2],\* and Yeol Je Cho [3,4]**

[1] College of Public Foundation, Yunnan Open University (Yunnan Technical College of National Defence Industry), Kunming 650500, China; kmszmzl@126.com

[2] Division of Research Administration, Yunnan University of Finance and Economics, Kunming 650221, China

[3] Department of Mathematics Education, Gyeongsang National University, Jinju 52828, Korea; yjcho@gnu.ac.kr

[4] School of Mathematical Science, University of Electronic Science and Technology of China, Chengdu 611731, China

\* Correspondence: WL64mail@aliyun.com; Tel.: +86-0871-6839-1896

**Abstract:** In this paper, a new algorithm for finding a common element of a split equality fixed point problem for nonexpansive mappings and split equality equilibrium problem in three Banach spaces is introduced. Also, some strong and weak convergence theorems for the proposed algorithm are proved. Finally, the main results obtained in this paper are applied to solve the split equality convex minimization problem.

**Keywords:** split equality equilibrium problem; split equality convex minimization problem; nonexpansive mapping; Banach space

## 1. Introduction

Let $\mathcal{C}$ be a closed convex subset of real Banach space $\mathcal{E}$ with the dual space $\mathcal{E}^*$, and let $f : \mathcal{C} \times \mathcal{C} \to \Re$ be a bifunction, where $\Re$ is the set of real numbers. The *equilibrium problem* (for short, (EP)) is to find $u^* \in \mathcal{C}$ such that:

$$f(u^*, u) \geq 0 \text{ for all } u \in \mathcal{C}.$$

The solutions set of the problem (EP) is denoted by $\text{EP}(f)$, that is,

$$\text{EP}(f) = \{u^* \in \mathcal{C} : f(u^*, u) \geq 0, u \in \mathcal{C}\}.$$

It is well known that many problems in physics, optimization, economics and other applied sciences reduce to find a solution of the problem (EP). Equilibrium problems and variational inequality problems in Hilbert spaces or Banach spaces have been extensively studied by many authors (see, for example, [1–8] and the references therein).

In order to model inverse problems in phase retrievals and medical image reconstruction [9], Censor and Elfving [10] introduced the following *split feasibility problem* (shortly, (SFP)) in 1994:

$$\text{Find } u^* \text{ such that } u^* \in \mathcal{C} \text{ and } g(u^*) \in \mathcal{Q},$$

where $\mathcal{C}$ and $\mathcal{Q}$ are nonempty closed convex subsets of Hilbert spaces $\mathcal{H}_1$ and $\mathcal{H}_2$, respectively, $g : \mathcal{H}_1 \to \mathcal{H}_2$ is a bounded linear operator.

As a matter of fact, many problems appeared in image restoration, computer tomograph and radiation therapy treatment planing can be formulated as the problem (SFP) [11–13]. For approximating solutions of the problem (SFP), some methods have been proposed by some authors (see, for instance [9,14–17].

Further, Moudafi [18] developed the problem (SFP) and proposed the split equality problem as follows:

Let $\mathcal{C}$, $\mathcal{Q}$ be two nonempty closed convex subsets of real Hilbert spaces $\mathcal{H}_1$ and $\mathcal{H}_2$, respectively, $\mathcal{H}_3$ be a real Hilbert space, $g : \mathcal{H}_1 \to \mathcal{H}_3$ and $h : \mathcal{H}_2 \to \mathcal{H}_3$ be two bounded linear operators. The *split equality problem* (shortly, (SEP)) is as follows:

$$\text{Find } u^* \in \mathcal{C} \text{ and } v^* \in \mathcal{Q} \text{ such that } g(u^*) = h(v^*).$$

It is easy to see that the problem (SEP) may reduce to the problem (SFP) when $\mathcal{H}_2 = \mathcal{H}_3$ and $h$ is the identity mapping $\mathcal{I}$ on $\mathcal{H}_2$. If $\mathcal{C}$ and $\mathcal{Q}$ are the sets of nonempty fixed points of the mappings $\mathcal{T}$ and $\mathcal{S}$ on $\mathcal{H}_1$ and $\mathcal{H}_2$, respectively, then, the split equality problem is called the *split equality fixed point problem* (shortly, (SEFP)) [19]). The set of solutions of the problem (SEFP) on $\mathcal{T}$ and $\mathcal{S}$ is denoted as follows:

$$\text{SEFP}(\mathcal{T}, \mathcal{S}) = \{(u^*, v^*) \in \mathcal{C} \times \mathcal{Q} : u^* \in Fix(\mathcal{T}), \ v^* \in Fix(\mathcal{S}), \ g(u^*) = h(v^*)\}.$$

Based on the idea of the split feasibility problem, in 2013, Kazmi and Rizvi [20] proposed the split equilibrium problems in Hilbert spaces.

Assume that $f_1 : \mathcal{C} \times \mathcal{C} \to \Re$ and $f_2 : \mathcal{Q} \times \mathcal{Q} \to \Re$ are nonlinear bifunctions, where $\mathcal{C}$ and $\mathcal{Q}$ are closed convex subsets of $\mathcal{H}_1$ and $\mathcal{H}_2$, respectively, and $g : \mathcal{H}_1 \to \mathcal{H}_2$ is a bounded linear operator. The *split equilibrium problem* (shortly, (SEQP)) is as follows:

$$\text{Find } u^* \in \mathcal{C} \text{ such that } f_1(u^*, u) \geq 0 \text{ for all } u \in \mathcal{C}$$

and such that

$$v^* = g(u^*) \in \mathcal{Q} \text{ solves } f_2(v^*, v) \geq 0 \text{ for all } v \in \mathcal{Q}.$$

Here, based on the ideas of the problems (SEP) and (SEQP), we consider the following so-called split equality equilibrium problem in Banach spaces:

**Definition 1.** *Let $\mathcal{E}_1$, $\mathcal{E}_2$, $\mathcal{E}_3$ be three Banach spaces and $\mathcal{C}$, $\mathcal{Q}$ be nonempty closed convex subsets of $\mathcal{E}_1$, $\mathcal{E}_2$, respectively. Let $f_1 : \mathcal{E}_1 \times \mathcal{E}_1 \to \Re$, $f_2 : \mathcal{E}_2 \times \mathcal{E}_2 \to \Re$ be two bifunctions and $g : \mathcal{E}_1 \to \mathcal{E}_3$, $h : \mathcal{E}_2 \to \mathcal{E}_3$ be two bounded linear operators. The split equality equilibrium problem (shortly, (SEEP)) is as follows: Find $u^* \in \mathcal{C}$ and $v^* \in \mathcal{Q}$ such that*

$$f_1(u^*, u) \geq 0, \ f_2(v^*, v) \geq 0 \text{ for all } u \in \mathcal{C}, \ v \in \mathcal{Q} \text{ and } g(u^*) = h(v^*).$$

The set of solutions of the problem (SEEP) is denoted by $\text{SEEP}(f_1, f_2)$, that is,

$$\begin{aligned}
&\text{SEEP}(f_1, f_2) \\
&= \{(u^*, v^*) \in \mathcal{C} \times \mathcal{Q} : f_1(u^*, u) \geq 0, \ f_2(v^*, v) \geq 0, \ u \in \mathcal{C}, \ v \in \mathcal{Q}, \ g(u^*) = h(v^*)\}.
\end{aligned}$$

The problems (SFP), (SEQP) and (SEP) in Hilbert spaces have attracted the attention of many authors. Some iteration algorithms have been proposed for finding a solution of these problems (see, for instance, ref. [20–23] and the references therein). Especially, the split equality mixed equilibrium problem was investigated in [24] and the convergence results on solutions were obtained in Hilbert spaces.

The recent research on the probelms (SFP), (SEP), the split common null point problem and the split common fixed point problem have been developed in Banach spaces by some authors (see, for

example, ref. [25–27] and the references therein). But, according to the literature, we can not find out the results on the problems (SEP) and (SEQP) in Banach spaces.

In this paper, motivated and inspired by the recent works in [20,23–27]), we construct a new algorithm to find a common element of the problem (SEFP) and the problem (SEEP) for nonexpansive mappings in three Banach spaces. Also, some strong and weak convergence theorems for the proposed algorithm are proved. Finally, our main results are applied to study the convergence of solutions of a split equality convex minimization problem.

## 2. Preliminaries

In this paper, we denote the strong convergence and weak convergence of a sequence $\{x_n\}$ to a point $x \in \mathcal{E}$ by $x_n \to x$ and $x_n \rightharpoonup x$, respectively.

Let $\mathcal{E}$ be a real normed linear space and $\mathcal{C}$ be a nonempty closed convex subset of $\mathcal{E}$. A mapping $\mathcal{T} : \mathcal{C} \to \mathcal{C}$ is said to be *nonexpansive* if

$$\|\mathcal{T}x - \mathcal{T}y\| \le \|x - y\| \text{ for all } x, y \in \mathcal{C}.$$

If $\mathcal{C}$ is a bounded closed convex subset of a uniformly convex Banach space $\mathcal{E}$ and $\mathcal{T} : \mathcal{C} \longrightarrow \mathcal{C}$ is the nonexpansive, then the fixed point set $Fix(\mathcal{T})$ is nonempty (see [28] for more details).

Let $\mathcal{E}$ be a real Banach space with the dual space $\mathcal{E}^*$. The *normalized duality mapping* $\mathcal{J}$ from $\mathcal{E}$ to $2^{\mathcal{E}^*}$ is defined by

$$\mathcal{J}(x) = \{x^* \in \mathcal{E}^* : \langle x, x^* \rangle = \|x\|^2 = \|x^*\|^2\} \text{ for all } x \in \mathcal{E},$$

where $\langle \cdot, \cdot \rangle$ denotes the generalized duality pairing between $\mathcal{E}$ and $\mathcal{E}^*$.

Note that, by the Hahn-Banach theorem, $\mathcal{J}(x)$ is nonempty (see [28] for more details) and, if $\mathcal{E} := \mathcal{H}$ is a Hilbert space, then $\mathcal{J}$ is the identity mapping on $\mathcal{E}$.

**Proposition 1** ([28,29]). *Assume that $\mathcal{E}$ is a Banach space and $\mathcal{J}$ is the normalized duality mapping from $\mathcal{E}$ into $\mathcal{E}^*$. Then*

1. *If $\mathcal{E}$ is reflexive, strictly convex and smooth Banach space, then $\mathcal{J}$ is single-valued, one-to-one and surjective, and $\mathcal{J}^{-1} : \mathcal{E}^* \to \mathcal{E}$ is the inverse of $\mathcal{J}$.*
2. *If $\mathcal{E}$ is a uniformly smooth Banach spaces, then $\mathcal{J}$ is uniformly norm-to-norm continuous on each bounded subset of $\mathcal{E}$.*

The normalized duality mapping $\mathcal{J}$ is said to be *weakly sequentially continuous* if the weak convergence of a sequence $\{x_n\}$ to $x \in \mathcal{E}$ implies the weak* convergence of $\{\mathcal{J}(x_n)\}$ to $\mathcal{J}(x)$ in $\mathcal{E}^*$.

**Definition 2.** *Let $\mathcal{C}$ be a nonempty closed convex subset of a Banach space $\mathcal{E}$. The mapping $\mathcal{A} : \mathcal{C} \to \mathcal{E}$ is said to be:*

1. *accretive if*
$$\langle \mathcal{A}x - \mathcal{A}y, \mathcal{J}(x - y) \rangle \ge 0 \text{ for all } x, y \in \mathcal{C};$$

2. *strongly accretive if there exists a constant $c > 0$ such that*

$$\langle \mathcal{A}x - \mathcal{A}y, \mathcal{J}(x - y) \rangle \ge c\|x - y\|^2 \text{ for all } x, y \in \mathcal{C};$$

3. *$\alpha$-inverse strongly accretive if there exists a constant $\alpha > 0$ such that*

$$\langle \mathcal{A}x - \mathcal{A}y, \mathcal{J}(x - y) \rangle \ge \alpha\|\mathcal{A}x - \mathcal{A}y\|^2 \text{ for all } x, y \in \mathcal{C}.$$

For solving the equilibrium problem, we assume that the bifunction $f : \mathcal{C} \times \mathcal{C} \to \Re$ satisfies the following conditions:

(C1)  $f(x, x) = 0$ for all $x \in \mathcal{C}$;
(C2)  $f(x, y) + f(y, x) \leq 0$ for all $x, y \in \mathcal{C}$;
(C3)  for all $x, y, z \in \mathcal{C}$, $\lim_{t \downarrow 0} f(tz + (1 - t)x, y) \leq f(x, y)$;
(C4)  for all $x \in \mathcal{C}$, the function $y \longmapsto f(x, y)$ is convex and lower semi-continuous.

**Lemma 1** ([2]). *Let $\mathcal{C}$ be a closed convex subset of a smooth, strictly convex and reflexive Banach space $\mathcal{E}$ and $f : \mathcal{C} \times \mathcal{C} \to \mathfrak{R}$ be a bifunction satisfying (C1)–(C4). For any $r > 0$ and $x \in \mathcal{E}$, there exists $z \in \mathcal{C}$ such that*

$$f(z, y) + \frac{1}{r} \langle y - z, \mathcal{J}z - \mathcal{J}x \rangle \geq 0 \text{ for all } y \in \mathcal{C}.$$

**Lemma 2** ([30]). *Let $\mathcal{C}$ be a closed convex subset of a smooth, strictly convex and reflexive Banach space $\mathcal{E}$ and $f : \mathcal{C} \times \mathcal{C} \to \mathfrak{R}$ be a bifunction satisfying (C1)–(C4). For any $r > 0$ and $x \in \mathcal{E}$, define a mapping $\mathcal{T}_r^{\mathcal{F}} : \mathcal{C} \to \mathcal{C}$ as follows:*

$$\mathcal{T}_r^{\mathcal{F}}(x) = \{z \in \mathcal{C} : f(z, y) + \frac{1}{r} \langle y - z, \mathcal{J}z - \mathcal{J}x \rangle \geq 0, y \in \mathcal{C}\},$$

*Then the following hold:*

1.  $\mathcal{T}_r^{\mathcal{F}}$ *is a singleton;*
2.  $\mathcal{T}_r^{\mathcal{F}}$ *is firmly nonexpansive, that is, for all $u, v \in E$,*

$$\langle \mathcal{T}_r^{\mathcal{F}} u - \mathcal{T}_r^{\mathcal{F}} v, \mathcal{J}\mathcal{T}_r^{\mathcal{F}} u - \mathcal{J}\mathcal{T}_r^{\mathcal{F}} v \rangle \leq \langle \mathcal{T}_r^{\mathcal{F}} u - \mathcal{T}_r^{\mathcal{F}} v, \mathcal{J}u - \mathcal{J}v \rangle;$$

3.  $Fix(\mathcal{T}_r^{\mathcal{F}}) = \mathrm{EP}(f)$;
4.  $\mathrm{EP}(f)$ *is closed and convex.*

**Lemma 3** ([31]). *For any number $r > 0$, a real Banach space $\mathcal{E}$ is uniformly convex if and only if there exists a continuous strictly increasing function $g : [0, \infty) \to [0, \infty)$ with $g(0) = 0$ such that*

$$\|tu + (1 - t)v\|^2 \leq t\|u\|^2 + (1 - t)\|v\|^2 - t(1 - t)g(\|u - v\|)$$

*for all $u, v \in \mathcal{E}$ with $\|u\| \leq r$ and $\|v\| \leq r$ and $t \in [0, 1]$.*

Let $\mathcal{T} : \mathcal{C} \to \mathcal{C}$ be a mapping with $Fix(\mathcal{T}) \neq \emptyset$. $\mathcal{T}$ is said to be *demi-closed* at zero if, for any $\{x_n\} \subset \mathcal{C}$ with $x_n \rightharpoonup x$ and $\|x_n - \mathcal{T}x_n\| \to 0$, then $x = \mathcal{T}x$. A mapping $\mathcal{T} : \mathcal{C} \to \mathcal{C}$ is said to be *semi-compact* if for any bounded sequence $\{x_n\}$ in $\mathcal{C}$ such that $\|x_n - \mathcal{T}x_n\| \to 0, (n \to \infty)$, there exists a subsequence $\{x_{n_j}\}$ of $\{x_n\}$ such that $\{x_{n_j}\}$ converges strongly to $x^* \in \mathcal{C}$.

A Banach space $E$ is said to satisfy *Opial's property* if, for any sequence $\{x_n\}$ in $E$ with $x_n \rightharpoonup x$, for any $y \in \mathcal{E}$ with $y \neq x$, we have

$$\liminf_{n \to \infty} \|x_n - x\| < \liminf_{n \to \infty} \|x_n - y\|.$$

**Lemma 4** ([31]). *Let $\mathcal{E}$ be a 2-uniformly smooth Banach space with the best smoothness constants $\mathcal{K} > 0$. Then the following inequality holds:*

$$\|x + y\|^2 \leq \|x\|^2 + 2\langle y, \mathcal{J}x \rangle + 2\|\mathcal{K}y\|^2 \text{ for all } x, y \in \mathcal{E}.$$

**Lemma 5** ([32]). *Let $\mathcal{C}$ be a nonempty closed subset of a real uniformly convex Banach space $\mathcal{E}$ and $\mathcal{T} : \mathcal{C} \to \mathcal{C}$ be a nonexpansive mapping. Then $\mathcal{T}$ is demi-closed at zero.*

## 3. Main Results

Throughout the rest of this paper, we always assume the following conditions are satisfied:

(*A*)  $\mathcal{E}_1$, $\mathcal{E}_2$ are real uniformly convex and 2-uniformly smooth Banach spaces satisfying Opial's condition and with the best smoothness constant $k$ satisfying $0 < k \leq \frac{1}{\sqrt{2}}$;

(*B*)  $\mathcal{E}_3$ is a smooth, reflexive and strictly convex Banach space;

(*C*)  $f_1 : \mathcal{E}_1 \times \mathcal{E}_1 \to \Re$ and $f_2 : \mathcal{E}_2 \times \mathcal{E}_2 \to \Re$ are the bifunctions satisfying the conditions (C1)–(C4);

(*D*)  $\mathcal{T} : \mathcal{E}_1 \to \mathcal{E}_1$, $\mathcal{S} : \mathcal{E}_2 \to \mathcal{E}_2$ are two nonexpansive mappings with $Fix(\mathcal{T}) \neq \emptyset$ and $Fix(\mathcal{S}) \neq \emptyset$;

(*E*)  $g : \mathcal{E}_1 \to \mathcal{E}_3$, $h : \mathcal{E}_2 \to \mathcal{E}_3$ are two bounded linear operators with adjoints $g^*$, $h^*$, respectively.

**Theorem 1.** *Let $\mathcal{E}_1$, $\mathcal{E}_2$, $\mathcal{E}_3$ $f_1$, $f_2$, $\mathcal{T}$, $\mathcal{S}$, $g$ and $h$ be the same as above. Let $\{(x_n, y_n)\}$ be the iteration scheme in $\mathcal{E}_1 \times \mathcal{E}_2$ defined as follows: for any $(x_1, y_1) \in \mathcal{E}_1 \times \mathcal{E}_2$,*

$$
\begin{cases}
f_1(u_n, u) + \frac{1}{r}\langle u - u_n, \mathcal{J}_1 u_n - \mathcal{J}_1 x_n \rangle \geq 0, \ \forall u \in \mathcal{E}_1, \\
f_2(v_n, v) + \frac{1}{r}\langle v - v_n, \mathcal{J}_2 v_n - \mathcal{J}_2 y_n \rangle \geq 0, \ \forall v \in \mathcal{E}_2, \\
x_{n+1} = \alpha_n x_n + (1 - \alpha_n)\mathcal{T}(u_n - \rho \mathcal{J}_1^{-1} g^* \mathcal{J}_3(g(u_n) - h(v_n))), \\
y_{n+1} = \alpha_n y_n + (1 - \alpha_n)\mathcal{S}(v_n + \rho \mathcal{J}_2^{-1} h^* \mathcal{J}_3(g(u_n) - h(v_n))), \ \forall n \geq 1,
\end{cases}
\tag{1}
$$

*where $r \in (0, \infty)$, $(\|h\|^2 + \|g\|^2)^{-1} < \rho < 2(\|h\|^2 + \|g\|^2)^{-1}$ and $\{\alpha_n\}$ is a sequence in $[a, b]$ for some $a, b \in (0, 1)$.*

*If $\Gamma := \mathrm{SEFP}(\mathcal{T}, \mathcal{S}) \bigcap \mathrm{SEEP}(f_1, f_2) \neq \emptyset$, then we have the following:*

1.  *$\{(x_n, y_n)\} \rightharpoonup (p, q) \in \Gamma$;*
2.  *Furthermore, if $\mathcal{S}$ and $\mathcal{T}$ are semi-compact, then $\{(x_n, y_n)\} \to (p, q) \in \Gamma$.*

**Proof.** Since $\mathcal{E}_1$, $\mathcal{E}_2$ are real uniformly convex and 2-uniformly smooth Banach spaces, $\mathcal{E}_3$ is a smooth, reflexive and strictly convex Banach space, by the properties of the the normalized duality mapping $\mathcal{J}$, we know that the iteration scheme (1) is well defined.

1.  For 1, we divide the proof of the Conclusion 1 into four steps as follows:

**Step 1.** Show that the limit of the sequence $\{\|x_{n+1} - x\|^2 + \|y_{n+1} - y\|\}^2$ exists for any $(x, y) \in \Gamma$. In fact, taking $(x, y) \in \Gamma$, from Lemma 2, we know that $x = \mathcal{T}_r^{f_1} x$ and $y = \mathcal{T}_r^{f_2} y$. Furthermore, we have

$$
\|u_n - x\| = \|\mathcal{T}_r^{f_1} x_n - \mathcal{T}_r^{f_1} x\| \leq \|x_n - x\|
\tag{2}
$$

and

$$
\|v_n - y\| = \|\mathcal{T}_r^{f_2} y_n - \mathcal{T}_r^{f_2} y\| \leq \|y_n - y\|.
\tag{3}
$$

Because of the nonexpansiveness of $\mathcal{S}$ and $\mathcal{T}$, using (2), (3), Lemma 3 and Lemma 4, we have

$$
\begin{aligned}
\|x_{n+1} - x\|^2 &= \|\alpha_n x_n + (1 - \alpha_n)\mathcal{T}(u_n - \rho \mathcal{J}_1^{-1} g^* \mathcal{J}_3(g(u_n) - h(v_n))) - x\|^2 \\
&\leq \alpha_n \|x_n - x\|^2 + (1 - \alpha_n)\|\mathcal{T}(u_n - \rho \mathcal{J}_1^{-1} g^* \mathcal{J}_3(g(u_n) - h(v_n))) - x\|^2 \\
&\quad - \alpha_n(1 - \alpha_n)g_1(\|x_n - \mathcal{T}(u_n - \rho \mathcal{J}_1^{-1} g^* \mathcal{J}_3(g(u_n) - h(v_n)))\|) \\
&\leq \alpha_n \|x_n - x\|^2 + (1 - \alpha_n)\|u_n - \rho \mathcal{J}_1^{-1} g^* \mathcal{J}_3(g(u_n) - h(v_n)) - x\|^2 \\
&\quad - \alpha_n(1 - \alpha_n)g_1(\|x_n - \mathcal{T}(u_n - \rho \mathcal{J}_1^{-1} g^* \mathcal{J}_3(g(u_n) - h(v_n)))\|) \\
&\leq \alpha_n \|x_n - x\|^2 + (1 - \alpha_n)[\|\rho \mathcal{J}_1^{-1} g^* \mathcal{J}_3(g(u_n) - h(v_n))\|^2 \\
&\quad + 2\rho\langle x - u_n, \mathcal{J}_1 \mathcal{J}_1^{-1} g^* \mathcal{J}_3(g(u_n) - h(v_n))\rangle + 2k^2\|u_n - x\|^2] \\
&\quad - \alpha_n(1 - \alpha_n)g_1(\|x_n - \mathcal{T}(u_n - \rho \mathcal{J}_1^{-1} g^* \mathcal{J}_3(g(u_n) - h(v_n)))\|) \\
&\leq \alpha_n \|x_n - x\|^2 + (1 - \alpha_n)[\rho^2\|g\|^2\|g(u_n) - h(v_n)\|^2 \\
&\quad + 2\rho\langle g(x) - g(u_n), \mathcal{J}_3(g(u_n) - h(v_n))\rangle + 2k^2\|u_n - x\|^2] \\
&\quad - \alpha_n(1 - \alpha_n)g_1(\|x_n - \mathcal{T}(u_n - \rho \mathcal{J}_1^{-1} g^* \mathcal{J}_3(g(u_n) - h(v_n)))\|) \\
&\leq [\alpha_n + 2k^2(1 - \alpha_n)]\|x_n - x\|^2 + (1 - \alpha_n)\rho^2\|g\|^2\|g(u_n) - h(v_n)\|^2 \\
&\quad + 2(1 - \alpha_n)\rho\langle g(x) - g(u_n), \mathcal{J}_3(g(u_n) - h(v_n))\rangle \\
&\quad - \alpha_n(1 - \alpha_n)g_1(\|x_n - \mathcal{T}(u_n - \rho \mathcal{J}_1^{-1} g^* \mathcal{J}_3(g(u_n) - h(v_n)))\|) \\
&\leq \|x_n - x\|^2 + (1 - \alpha_n)\rho^2\|g\|^2\|g(u_n) - h(v_n)\|^2 \\
&\quad + 2(1 - \alpha_n)\rho\langle g(x) - g(u_n), \mathcal{J}_3(g(u_n) - h(v_n))\rangle - \alpha_n(1 - \alpha_n)g_1(\|x_n - \mathcal{T}z_n\|),
\end{aligned}
\tag{4}
$$

where $z_n = u_n - \rho \mathcal{J}_1^{-1} g^* \mathcal{J}_3(g(u_n) - h(v_n))$. Setting $e_n = v_n + \rho \mathcal{J}_2^{-1} h^* \mathcal{J}_3(g(u_n) - h(v_n))$, it follows from (1) that

$$
\begin{aligned}
\|y_{n+1} - y\|^2 &\leq \|y_n - y\|^2 + (1 - \alpha_n)\rho^2\|h\|^2\|g(u_n) - h(v_n)\|^2 \\
&\quad + 2(1 - \alpha_n)\rho\langle h(v_n) - h(y), \mathcal{J}_3(g(u_n) - h(v_n))\rangle - \alpha_n(1 - \alpha_n)g_2(\|y_n - \mathcal{S}e_n\|).
\end{aligned}
\tag{5}
$$

Since $(x, y) \in \Gamma$, we know that $g(x) = h(y)$ and so, by (4) and (5),

$$
\begin{aligned}
&\|x_{n+1} - x\|^2 + \|y_{n+1} - y\|^2 \\
&\leq (\|x_n - x\|^2 + \|y_n - y\|^2) + (1 - \alpha_n)\rho^2(\|g\|^2 + \|h\|^2)\|g(u_n) - h(v_n)\|^2 \\
&\quad + 2(1 - \alpha_n)\rho\langle h(v_n) - g(u_n), \mathcal{J}_3(g(u_n) - h(v_n))\rangle \\
&\quad - \alpha_n(1 - \alpha_n)[g_1(\|u_n - \mathcal{T}z_n\|) + g_2(\|v_n - \mathcal{S}e_n\|)] \\
&\leq (\|x_n - x\|^2 + \|y_n - y\|^2) - (1 - \alpha_n)\rho[2 - (\|g\|^2 + \|h\|^2)\rho]\|g(u_n) - h(v_n)\|^2 \\
&\quad - \alpha_n(1 - \alpha_n)[g_1(\|x_n - \mathcal{T}z_n\|) + g_2(\|y_n - \mathcal{S}e_n\|)].
\end{aligned}
\tag{6}
$$

Let $\Gamma_n(x, y) := \|x_n - x\|^2 + \|y_n - y\|^2$. Then, by (6), we have

$$
\begin{aligned}
\Gamma_{n+1}(x, y) &\leq \Gamma_n(x, y) - (1 - \alpha_n)\rho[2 - (\|g\|^2 + \|h\|^2)\rho]\|g(u_n) - h(v_n)\|^2 \\
&\quad - \alpha_n(1 - \alpha_n)[g_1(\|x_n - \mathcal{T}z_n\|) + g_2(\|y_n - \mathcal{S}e_n\|)]
\end{aligned}
\tag{7}
$$

Since $0 < k \leq \frac{1}{\sqrt{2}}$ and $(\|g\|^2 + \|h\|^2)^{-1} < \rho < 2(\|g\|^2 + \|h\|^2)^{-1}$, we have $0 < 2 - \rho(\|g\|^2 + \|h\|^2) < 1$ and so, from (7), it follows that $\Gamma_n(x, y) = \|x_n - x\|^2 + \|y_n - y\|^2$ is decreasing. So, $\lim_{n \to \infty} \Gamma_n(x, y)$ exists. Further, it is easy to see that $\{x_n\}$ and $\{y_n\}$ are bounded.

**Step 2.** Show that

$$
\lim_{n \to \infty} \|g(u_n) - h(v_n)\| = 0, \quad \lim_{n \to \infty} \|x_n - u_n\| = 0, \quad \lim_{n \to \infty} \|y_n - v_n\| = 0.
$$

In fact, it follows from (7) that

$$
\begin{aligned}
(1 - \alpha_n)\rho[2 - (\|g\|^2 + \|h\|^2)\rho]&\|g(u_n) - h(v_n)\|^2 \\
&+ \alpha_n(1 - \alpha_n)[g_1(\|x_n - \mathcal{T}z_n\|) + g_2(\|y_n - \mathcal{S}e_n\|)] \\
&\le \Gamma_n(x, y) - \Gamma_{n+1}(x, y).
\end{aligned} \tag{8}
$$

Since $(\|g\|^2 + \|h\|^2)^{-1} < \rho < 2(\|g\|^2 + \|h\|^2)^{-1}$ and $\{\alpha_n\}$ is a sequence in $[a, b]$ for some $a, b \in (0, 1)$, by (8), we have

$$
\lim_{n\to\infty} g_1(\|x_n - \mathcal{T}z_n\|) = 0, \quad \lim_{n\to\infty} g_2(\|y_n - \mathcal{S}e_n\|) = 0 \tag{9}
$$

and

$$
\lim_{n\to\infty} \|g(u_n) - h(v_n)\| = 0. \tag{10}
$$

Applying the properties of $g_1$, $g_2$, (9) and Lemma 3, we have

$$
\lim_{n\to\infty} \|x_n - \mathcal{T}z_n\| = 0, \quad \lim_{n\to\infty} \|y_n - \mathcal{S}e_n\| = 0. \tag{11}
$$

Since

$$
\|u_n - z_n\| = \|\mathcal{J}_1(u_n - z_n)\| = \|\rho g^* \mathcal{J}_3(g(u_n) - h(v_n))\| \le \rho\|g\|\|g(u_n) - h(v_n)\|
$$

and

$$
\|v_n - e_n\| = \|\mathcal{J}_2(v_n - e_n)\| = \|\rho h^* \mathcal{J}_3(g(u_n) - h(v_n))\| \le \rho\|h\|\|g(u_n) - h(v_n)\|,
$$

it follows from (10) that

$$
\lim_{n\to\infty} \|u_n - z_n\| = 0, \quad \lim_{n\to\infty} \|v_n - e_n\| = 0.
$$

In addition, since

$$
\|x_{n+1} - x_n\| = \|\alpha_n x_n + (1 - \alpha_n)\mathcal{T}z_n - x_n\| = (1 - \alpha_n)\|\mathcal{T}z_n - x_n\|,
$$

by (11), we have

$$
\lim_{n\to\infty} \|x_{n+1} - x_n\| = 0. \tag{12}
$$

Similarly, we obtain

$$
\lim_{n\to\infty} \|y_{n+1} - y_n\| = 0. \tag{13}
$$

Again, since

$$
\|u_{n+1} - u_n\| = \|\mathcal{T}_r^{f_1} x_{n+1} - \mathcal{T}_r^{f_1} x_n\| \le \|x_{n+1} - x_n\|
$$

and

$$
\|v_{n+1} - v_n\| = \|\mathcal{T}_r^{f_2} y_{n+1} - \mathcal{T}_r^{f_2} y_n\| \le \|y_{n+1} - y_n\|,
$$

by (12) and (13), it follows that

$$
\lim_{n\to\infty} \|u_{n+1} - u_n\| = 0, \quad \lim_{n\to\infty} \|v_{n+1} - v_n\| = 0. \tag{14}
$$

Since $(x, y) \in \Gamma$, we have $x = \mathcal{T}_r^{f_1} x$ and $y = \mathcal{T}_r^{f_2} y$. In addition, it follows from Lemma 2 that $\mathcal{T}_r$ is firmly nonexpansive. Further, we have

$$
\begin{aligned}
\|u_n - x\|^2 &= \|\mathcal{T}_r^{f_1} x_n - \mathcal{T}_r^{f_1} x\|^2 \\
&= \langle \mathcal{T}_r^{f_1} x_n - \mathcal{T}_r^{f_1} x, \mathcal{J}(\mathcal{T}_r^{f_1} x_n - \mathcal{T}_r^{f_1} x) \rangle \\
&\leq \|u_n - x\| \|\mathcal{T}_r^{f_1} x_n - \mathcal{T}_r^{f_1} x\| \\
&\leq \|u_n - x\| \|x_n - x\| \\
&\leq \frac{1}{2}(\|x_n - x\|^2 + \|u_n - x\|^2 - \|x_n - u_n\|^2)
\end{aligned}
\tag{15}
$$

and

$$
\begin{aligned}
\|v_n - y\|^2 &= \|\mathcal{T}_r^{f_2} y_n - \mathcal{T}_r^{f_2} y\|^2 \\
&\leq \frac{1}{2}(\|y_n - y\|^2 + \|v_n - y\|^2 - \|y_n - v_n\|^2).
\end{aligned}
\tag{16}
$$

So, it follows from (15) and (16) that

$$
\|u_n - x\|^2 \leq \|x_n - x\|^2 - \|x_n - u_n\|^2, \quad \|v_n - y\|^2 \leq \|y_n - y\|^2 - \|y_n - v_n\|^2.
\tag{17}
$$

Also, it follows from (1) and Lemma 4 that

$$
\begin{aligned}
\|x_{n+1} - x\|^2 &= \|\alpha_n x_n + (1 - \alpha_n)\mathcal{T}(u_n - \rho \mathcal{J}_1^{-1} g^* \mathcal{J}_3(g(u_n) - h(v_n))) - x\|^2 \\
&\leq \alpha_n \|x_n - x\|^2 + (1 - \alpha_n)[\rho^2 \|g\|^2 \|g(u_n) - h(v_n)\|^2 \\
&\quad + 2\rho \langle g(x) - g(u_n), \mathcal{J}_3(g(u_n) - h(v_n)) \rangle \\
&\quad + 2k^2 \|u_n - x\|^2] - \alpha_n(1 - \alpha_n)g_1(\|x_n - \mathcal{T}z_n\|) \\
&\leq \alpha_n \|x_n - x\|^2 + (1 - \alpha_n)[\rho^2 \|g\|^2 \|g(u_n) - h(v_n)\|^2 \\
&\quad + 2\rho \langle g(x) - g(u_n), \mathcal{J}_3(g(u_n) - h(v_n)) \rangle \\
&\quad + 2k^2(\|x_n - x\|^2 - \|x_n - u_n\|^2)] - \alpha_n(1 - \alpha_n)g_1(\|x_n - \mathcal{T}z_n\|)
\end{aligned}
\tag{18}
$$

and

$$
\begin{aligned}
\|y_{n+1} - y\|^2 &\leq \alpha_n \|y_n - y\|^2 + (1 - \alpha_n)[\rho^2 \|h\|^2 \|g(u_n) - h(v_n)\|^2 \\
&\quad + 2\rho \langle h(v_n) - h(y), \mathcal{J}_3(g(u_n) - h(v_n)) \rangle \\
&\quad + 2k^2 \|v_n - y\|^2] - \alpha_n(1 - \alpha_n)g_2(\|y_n - \mathcal{S}e_n\|) \\
&\leq \alpha_n \|y_n - y\|^2 + (1 - \alpha_n)[\rho^2 \|h\|^2 \|g(u_n) - h(v_n)\|^2 \\
&\quad + 2\rho \langle h(v_n) - h(y), \mathcal{J}_3(g(u_n) - h(v_n)) \rangle \\
&\quad + 2k^2(\|y_n - y\|^2 - \|y_n - v_n\|^2)] - \alpha_n(1 - \alpha_n)g_2(\|y_n - \mathcal{S}e_n\|).
\end{aligned}
\tag{19}
$$

Adding the inequalities (18), (19) and taking into account the fact that $g(x) = h(y)$, we obtain

$$
\begin{aligned}
\|x_{n+1} &- x\|^2 + \|y_{n+1} - y\|^2 \\
&\leq [\alpha_n + (1 - \alpha_n)2k^2][\|x_n - x\|^2 + \|y_n - y\|^2] \\
&\quad - (1 - \alpha_n)\rho[2 - (\|g\|^2 + \|h\|^2)\rho]\|g(u_n) - h(v_n)\|^2 \\
&\quad - (1 - \alpha_n)2k^2[\|u_n - x\|^2 + \|v_n - y\|^2] \\
&\quad - \alpha_n(1 - \alpha_n)[g_1(\|x_n - \mathcal{T}z_n\|) + g_2(\|y_n - \mathcal{S}e_n\|)] \\
&\leq \|x_n - x\|^2 + \|y_n - y\|^2 - (1 - \alpha_n)\rho[2 - (\|g\|^2 + \|h\|^2)\rho]\|g(u_n) - h(v_n)\|^2 \\
&\quad - (1 - \alpha_n)2k^2[\|u_n - x\|^2 + \|v_n - y\|^2] \\
&\quad - \alpha_n(1 - \alpha_n)[g_1(\|x_n - \mathcal{T}z_n\|) + g_2(\|y_n - \mathcal{S}e_n\|)],
\end{aligned}
$$

and so

$$
\begin{aligned}
&(1-\alpha_n)2k^2\|x_n-u_n\|^2 + (1-\alpha_n)2k^2\|y_n-v_n\|^2 \\
&\leq \Gamma_n(x,y) - \Gamma_{n+1}(x,y) - (1-\alpha_n)\rho[2 - (\|g\|^2 + \|h\|^2)\rho]\|g(u_n)-h(v_n)\|^2 \\
&\quad - \alpha_n(1-\alpha_n)[g_1(\|x_n-\mathcal{T}z_n\|) + g_2(\|y_n-\mathcal{S}e_n\|)] \\
&\leq \Gamma_n(x,y) - \Gamma_{n+1}(x,y).
\end{aligned}
\tag{20}
$$

Since $\lim_{n\to\infty}\Gamma_n(x,y)$ exists, by (20), we have

$$
\lim_{n\to\infty}\|x_n-u_n\| = 0, \quad \lim_{n\to\infty}\|y_n-v_n\| = 0.
\tag{21}
$$

**Step 3.** Show that $\lim_{n\to\infty}\|x_n-\mathcal{T}x_n\| = 0$ and $\lim_{n\to\infty}\|y_n-\mathcal{S}y_n\| = 0$. In fact, using the nonexpansiveness of $\mathcal{T}$ and $\mathcal{S}$, we have

$$
\begin{aligned}
\|u_n-\mathcal{T}u_n\| &= \|u_n-x_{n+1}+x_{n+1}-\mathcal{T}u_n\| \\
&\leq \|u_n-x_{n+1}\| + \|x_{n+1}-\mathcal{T}u_n\| \\
&= \|u_n-u_{n+1}-u_{n+1}-x_{n+1}\| \\
&\quad + \|\alpha_n u_n + (1-\alpha_n)\mathcal{T}(x_n-\rho\mathcal{J}_1^{-1}g^*\mathcal{J}_3(g(u_n)-h(v_n))) - \mathcal{T}u_n\| \\
&\leq \|u_n-u_{n+1}\| + \|u_{n+1}-x_{n+1}\| + \alpha_n\|u_n-\mathcal{T}u_n\| \\
&\quad + (1-\alpha_n)\|\mathcal{T}(x_n-\rho\mathcal{J}_1^{-1}g^*\mathcal{J}_3(g(u_n)-h(v_n))) - \mathcal{T}u_n\| \\
&\leq \|u_n-u_{n+1}\| + \|u_{n+1}-x_{n+1}\| + \alpha_n\|u_n-\mathcal{T}u_n\| \\
&\quad + (1-\alpha_n)\|x_n-u_n\| + (1-\alpha_n)\|-\rho\mathcal{J}_1^{-1}g^*\mathcal{J}_3(g(u_n)-h(v_n))\|
\end{aligned}
$$

and, further,

$$
\begin{aligned}
(1-\alpha_n)\|u_n-\mathcal{T}u_n\| &\leq \|u_n-u_{n+1}\| + \|u_{n+1}-x_{n+1}\| + (1-\alpha_n)\|x_n-u_n\| \\
&\quad + (1-\alpha_n)\|-\rho\mathcal{J}_1^{-1}g^*\mathcal{J}_3(g(u_n)-h(v_n))\|.
\end{aligned}
\tag{22}
$$

By (14), (21) and (22), we have

$$
\lim_{n\to\infty}\|\mathcal{T}u_n-u_n\| = 0.
\tag{23}
$$

Similarly, we have

$$
\lim_{n\to\infty}\|\mathcal{S}v_n-v_n\| = 0.
\tag{24}
$$

Since

$$
\begin{aligned}
\|x_n-\mathcal{T}x_n\| &= \|x_n-u_n+u_n-\mathcal{T}u_n+\mathcal{T}u_n-\mathcal{T}x_n\| \\
&\leq \|x_n-u_n\| + \|u_n-\mathcal{T}u_n\| + \|\mathcal{T}u_n-\mathcal{T}x_n\| \\
&\leq 2\|x_n-u_n\| + \|u_n-\mathcal{T}u_n\|,
\end{aligned}
$$

it follows from (21) and (23) that

$$
\lim_{n\to\infty}\|x_n-\mathcal{T}x_n\| = 0.
\tag{25}
$$

In addition, we have

$$
\begin{aligned}
\|y_n-\mathcal{S}y_n\| &\leq \|y_n-v_n\| + \|v_n-\mathcal{S}v_n\| + \|\mathcal{S}v_n-\mathcal{S}y_n\| \\
&\leq 2\|y_n-v_n\| + \|v_n-\mathcal{S}v_n\|,
\end{aligned}
$$

and so it follows from (21) and (24) that

$$
\lim_{n\to\infty}\|y_n-\mathcal{S}y_n\| = 0.
\tag{26}
$$

**Step 4.** Show that $\{(x_n, y_n)\}$ has the unique weak cluster points $(x^*, y^*) \in \Gamma$. In fact, since $\mathcal{E}_1$ and $\mathcal{E}_2$ are reflexive, $\{x_n\}$ and $\{y_n\}$ are bounded, we may assume that $\{(x_n, y_n)\}$ has a weak cluster points $(x^*, y^*)$. Since $\mathcal{S}$ and $\mathcal{T}$ are nonespansive, $\mathcal{T}$ and $\mathcal{S}$ are demiclosed and so, from Lemma 5, (25) and (26), it follows that $x^* \in F(\mathcal{T})$ and $y^* \in F(\mathcal{S})$.

Now, we show that $x^* \in EP(f_1)$ and $y^* \in EP(f_2)$. Without loss of generality, we may suppose that the subsequence $\{(x_{n_i}, y_{n_i})\}$ of $\{(x_n, y_n)\}$ converges weakly to $(x^*, y^*)$. Also, by (21), we know that $\{(u_n, v_n)\}$ converges weakly to $(x^*, y^*)$. Using the uniformly norm-to-norm continuity of $\mathcal{J}_1$, it follows from (21) that

$$\lim_{n \to \infty} \|\mathcal{J}_1 x_n - \mathcal{J}_1 u_n\| = 0.$$

Since $u_n = \mathcal{T}_r^{f_1} x_n$, we have

$$f_1(u_n, u) + \frac{1}{r} < u - u_n, \mathcal{J}_1 x_n - \mathcal{J}_1 u_n > \geq 0 \text{ for all } u \in \mathcal{E}_1.$$

From the condition (C2), we obtain

$$\|u - u_n\| \frac{\|\mathcal{J}_1 x_n - \mathcal{J}_1 u_n\|}{r} \geq \frac{1}{r} \langle u - u_n, \mathcal{J}_1 x_n - \mathcal{J}_1 u_n \rangle \geq -f_1(u_n, u) \geq f_1(u, u_n)$$

for all $u \in \mathcal{E}_1$. Taking the limit as $n \to \infty$ in the inequality above , it follows from the condition (C4) and $u_n \rightharpoonup x^*$ that $f_1(u, x^*) \leq 0$ for all $u \in \mathcal{E}_1$. Put $z_t = tu + (1 - t)x^*$ for all $t \in (0, 1]$ and $u \in \mathcal{E}_1$. Thus we have $z_t \in \mathcal{E}_1$ and $f_1(z_t, x^*) \leq 0$. Applying the conditions (C1) and (C4), it follows that

$$0 = f_1(z_t, z_t) \leq t f_1(z_t, u) + (1 - t) f_1(z_t, x^*) \leq t f_1(z_t, u),$$

that is, $f_1(z_t, u) \geq 0$. As $t \to 0$, from the condition (C3), it follows that

$$f_1(x^*, u) \geq 0$$

for all $u \in \mathcal{E}_1$. This means that $x^* \in EP(f_1)$. Following the same argument above, we also have $y^* \in EP(f_2)$. Since $g$ and $h$ are bounded linear operators, the point $g(x^*) - h(y^*)$ is a weak cluster point of $\{g(u_n) - h(v_n)\}$. Again, applying the weakly lower semi-continuous property of the norm and (10), we obtain

$$\|g(x^*) - h(y^*)\| \leq \liminf_{n \to \infty} \|g(u_n) - h(v_n)\| = 0$$

and so $g(x^*) = h(y^*)$. Therefore, we have $(x^*, y^*) \in \Gamma$.

Now, we show that $(x^*, y^*)$ is the unique weak cluster point of $\{(x_n, y_n)\}$. Suppose that there exists another subsequence $\{(x_{n_k}, y_{n_k})\}$ of $\{(x_n, y_n)\}$ such that $\{(x_{n_k}, y_{n_k})\}$ converges weakly to a point $(p, q)$ with $(p, q) \neq (x^*, y^*)$. It is easy to see that $(p, q) \in \Gamma$. By Opial's properties of $\mathcal{E}_1$ and $\mathcal{E}_2$, we obtain

$$\liminf_{i \to \infty} \|x_{n_i} - p\| < \liminf_{i \to \infty} \|x_{n_i} - x^*\| = \lim_{n \to \infty} \|x_n - x^*\|$$

$$= \liminf_{k \to \infty} \|x_{n_k} - x^*\| < \liminf_{k \to \infty} \|x_{n_k} - p\|$$

$$= \lim_{n \to \infty} \|x_n - p\| = \liminf_{i \to \infty} \|x_{n_i} - p\|$$

and

$$\liminf_{i \to \infty} \|y_{n_i} - q\| < \liminf_{i \to \infty} \|y_{n_i} - y^*\| = \lim_{n \to \infty} \|y_n - y^*\|$$

$$= \liminf_{k \to \infty} \|y_{n_k} - y^*\| < \liminf_{k \to \infty} \|y_{n_k} - q\|$$

$$= \lim_{n \to \infty} \|y_n - q\| = \liminf_{i \to \infty} \|y_{n_i} - q\|,$$

which are contradictions and so $(p, q) = (x^*, y^*)$. This completes the proof of the Conclusion 1.

2. Now, we prove the Conclusion 2. In fact, since $\mathcal{S}$ and $\mathcal{T}$ are semi-compact, $\{(x_n, y_n)\}$ is bounded, $\lim_{n\to\infty} ||x_n - \mathcal{T}x_n|| = 0$ and $\lim_{n\to\infty} ||y_n - \mathcal{T}y_n|| = 0$, there exists a subsequence $\{(x_{n_j}, y_{n_j})\}$ of $\{(x_n, y_n)\}$ such that $\{(x_{n_j}, y_{n_j})\} \to (u^*, v^*)$. Since $\{(x_n, y_n)\} \rightharpoonup (x^*, y^*)$, we know that $(u^*, v^*) = (x^*, y^*)$.

On the other hand, since $\lim_{n\to\infty} \Gamma_n(x, y)$ exists for any $(x, y) \in \Gamma$ and $x_{n_j} \to x^*$, $y_{n_j} \rightharpoonup y^*$, we know that $\lim_{j\to\infty} \Gamma_{n_j}(x^*, y^*) = 0$. From the conclusion 1, we know that $\lim_{n\to\infty} \Gamma_n(x^*, y^*)$ exists and so $\lim_{n\to\infty} \Gamma_n(x^*, y^*) = 0$. Due to $0 \leq ||x_n - x^*||^2 \leq \Gamma_n(x^*, y^*)$ and $0 \leq ||y_n - y^*||^2 \leq \Gamma_n(x^*, y^*)$, we can obtain that

$$\lim_{n\to\infty} ||x_n - x^*|| = 0, \quad \lim_{n\to\infty} ||y_n - y^*|| = 0.$$

This completes the proof. □

Let $\phi : \mathcal{E}_1 \to \Re$ be a proper lower semi-continuous and convex functions, $\psi : \mathcal{E}_1 \to \mathcal{E}_1^*$ be a continuous and $\beta$-inverse strongly accretive mapping. Define

$$\mathcal{H}(\xi, y) = f(\xi, y) + \langle \psi\xi, y - \xi \rangle + \phi(y) - \phi(\xi) \text{ for all } \xi \in \mathcal{E}_1.$$

We can see that $\mathcal{H}(\xi, y)$ also satisfies the conditions (C1)–(C4) if $f$ satisfies the conditions (C1)–(C4). So, the problem (EP) reduces to the problem: Find $\xi^* \in \mathcal{E}_1$ such that

$$f(\xi^*, y) + \langle \psi\xi^*, y - \xi^* \rangle + \phi(y) - \phi(\xi^*) \geq 0 \text{ for all } y \in \mathcal{E}_1,$$

which is also called the *generalized mixed equilibrium problem* (shortly, (GMEP)).

The set of solutions of the problem (GMEP) is denoted by GMEP$(f, \psi, \phi)$.

If $\psi = 0$ in the problem (GMEP), then the problem (GMEP) reduces to the following problem: Find $\xi^* \in \mathcal{E}_1$ such that

$$f(\xi^*, y) + \phi(y) - \phi(\xi^*) \geq 0 \text{ for all } y \in \mathcal{E}_1,$$

which is also called the *mixed equilibrium problem* (shortly, (MEP)). The set of solutions of the problem (MEP) is denoted by MEP$(f, \phi)$.

**Definition 3.** *Let $\mathcal{E}_1$, $\mathcal{E}_2$, $\mathcal{E}_3$ be three Banach spaces, $f_1 : \mathcal{E}_1 \times \mathcal{E}_1 \to \Re$, $f_2 : \mathcal{E}_2 \times \mathcal{E}_2 \to \Re$ be two nonlinear bifunctions, $\psi_1 : \mathcal{E}_1 \to \mathcal{E}_1^*$, $\psi_2 : \mathcal{E}_2 \to \mathcal{E}_2^*$ be continuous and $\beta_i$-inverse strongly accretive mapping $(i = 1, 2)$, $\phi : \mathcal{E}_1 \to \Re \cup \{+\infty\}$, $\varphi : \mathcal{E}_2 \to \Re \cup \{+\infty\}$ be proper lower semi-continuous and convex functions and $g : \mathcal{E}_1 \to \mathcal{E}_3$, $h : \mathcal{E}_2 \to \mathcal{E}_3$ be two bounded linear operators. Then the split equality generalized mixed equilibrium problem (shortly, (SEGMEP)) is as follows: Find $\xi^* \in \mathcal{E}_1$ and $y^* \in \mathcal{E}_2$ such that*

$$\begin{cases} f_1(\xi^*, \xi) + \langle \psi_1\xi^*, \xi - \xi^* \rangle + \phi(\xi) - \phi(\xi^*) \geq 0, & \forall \xi \in \mathcal{E}_1, \\ f_2(y^*, y) + \langle \psi_2 y^*, y - y^* \rangle + \varphi(y) - \varphi(y^*) \geq 0, & \forall y \in \mathcal{E}_2, \\ g(\xi^*) = h(y^*). \end{cases}$$

The set of solutions of the problem (SEGMEP) is denoted by SEGMEP$(f_1, f_2, \psi_1, \psi_2, \phi, \varphi)$, that is,

$$\text{SEGMEP}(f_1, f_2, \psi_1, \psi_2, \phi, \varphi)$$
$$= \{(\xi^*, y^*) \in \mathcal{E}_1 \times \mathcal{E}_2 : f_1(\xi^*, \xi) + \langle \psi_1\xi^*, \xi - \xi^* \rangle + \phi(\xi) - \phi(\xi^*) \geq 0, \ \xi \in \mathcal{E}_1,$$
$$f_2(y^*, y) + \langle \psi_2 y^*, y - y^* \rangle + \varphi(y) - \varphi(y^*) \geq 0, \ y \in \mathcal{E}_2, \ g(\xi^*) = h(y^*)\}.$$

Taking

$$H_1(\xi, u) = f_1(\xi, u) + \langle \psi_1\xi, u - \xi \rangle + \phi(u) - \phi(\xi) \text{ for all } u \in \mathcal{E}_1$$

and

$$H_2(y, v) = f_2(y, v) + \langle \psi_2 y, v - y \rangle + \varphi(v) - \varphi(y) \text{ for all } v \in \mathcal{E}_2,$$

we can directly obtain the following result from Theorem 1 when $f_1$ and $f_2$ satisfy the conditions (C1)–(C4):

**Corollary 1.** *Let $\mathcal{E}_1$, $\mathcal{E}_2$, $\mathcal{E}_3$ $f_1$, $f_2$, $\mathcal{T}$, $\mathcal{S}$, $g$ and $h$ be the same as above. Let $\psi_1$, $\psi_2$, $\phi$ and $\varphi$ be the same as in Definition 3. Let iteration scheme $\{(x_n, y_n)\}$ be defined as follows: for any $(x_1, y_1) \in \mathcal{E}_1 \times \mathcal{E}_2$,*

$$
\begin{cases}
f_1(u_n, u) + \langle \psi_1 u_n, u - u_n \rangle + \phi(u) - \phi(u_n) + \frac{1}{r}\langle u - u_n, \mathcal{J}_1 u_n - \mathcal{J}_1 x_n \rangle \geq 0, \ \forall u \in \mathcal{E}_1, \\
f_2(v_n, v) + \langle \psi_2 v_n, v - v_n \rangle + \varphi(v) - \varphi(v_n) + \frac{1}{r}\langle v - v_n, \mathcal{J}_2 v_n - \mathcal{J}_2 y_n \rangle \geq 0, \ \forall v \in \mathcal{E}_2, \\
x_{n+1} = \alpha_n x_n + (1 - \alpha_n)\mathcal{T}(u_n - \rho \mathcal{J}_1^{-1} g^* \mathcal{J}_3(g(u_n) - h(v_n))), \\
y_{n+1} = \alpha_n y_n + (1 - \alpha_n)\mathcal{S}(v_n + \rho \mathcal{J}_2^{-1} h^* \mathcal{J}_3(g(u_n) - h(v_n))), \ \forall n \geq 1,
\end{cases}
$$

*where $r \in (0, \infty)$, $(\|g\|^2 + \|h\|^2)^{-1} < \rho < 2(\|g\|^2 + \|h\|^2)^{-1}$ and $\{\alpha_n\}$ is a sequence in $[a, b]$ for some $a, b \in (0, 1)$.*
*If $\Gamma := \mathrm{SEFP}(\mathcal{T}, \mathcal{S}) \cap \mathrm{SEGMEP}(f_1, f_2, \psi_1, \psi_2, \phi, \varphi) \neq \emptyset$, then we have the following:*

1. *$\{(x_n, y_n)\} \rightharpoonup (p, q) \in \Gamma$;*
2. *Furthermore, if $\mathcal{S}$ and $\mathcal{T}$ are semi-compact, then $\{(x_n, y_n)\} \to (p, q) \in \Gamma$.*

In Definition 3, if $\psi_1 = 0$ and $\psi_2 = 0$, then the problem (SEGEMP) reduces to the following so called the *split equality mixed equilibrium problem* (shortly, (SEMEP)) as follows: Find $\xi^* \in \mathcal{E}_1$ and $y^* \in \mathcal{E}_2$ such that

$$
\begin{cases}
f_1(\xi^*, \xi) + \phi(\xi) - \phi(\xi^*) \geq 0, \ \forall \xi \in \mathcal{E}_1, \\
f_2(y^*, y) + \varphi(y) - \varphi(y^*) \geq 0, \ \forall y \in \mathcal{E}_2, \\
g(x^*) = h(y^*).
\end{cases}
$$

The set of solutions of the problem (SEMEP) is denoted by $\mathrm{SEMEP}(f_1, f_2, \phi, \varphi)$, that is,

$$
\mathrm{SEMEP}(f_1, f_2, \phi, \varphi) = \{(\xi^*, y^*) \in \mathcal{E}_1 \times \mathcal{E}_2 : f_1(\xi^*, \xi) + \phi(\xi) - \phi(\xi^*) \geq 0, \ \xi \in \mathcal{E}_1,
$$
$$
f_2(y^*, y) + \varphi(y) - \varphi(y^*) \geq 0, \ y \in \mathcal{E}_2, \ g(\xi^*) = h(y^*)\}.
$$

Taking $\psi_1 = 0$ and $\psi_2 = 0$ in Corollary 1, we can obtain the following result:

**Corollary 2.** *Let $\mathcal{E}_1$, $\mathcal{E}_2$, $\mathcal{E}_3$ $f_1$, $f_2$, $\mathcal{T}$, $\mathcal{S}$, $g$ and $h$ be the same as above. Let $\phi$ and $\varphi$ be the same as in Definition 3. Let $\{(x_n, y_n)\}$ be the iteration scheme in $\mathcal{E}_1 \times \mathcal{E}_2$ defined as follows: for any $(x_1, y_1) \in \mathcal{E}_1 \times \mathcal{E}_2$,*

$$
\begin{cases}
f_1(u_n, u) + \phi(u) - \phi(u_n) + \frac{1}{r}\langle u - u_n, \mathcal{J}_1 u_n - \mathcal{J}_1 x_n \rangle \geq 0, \ \forall u \in \mathcal{E}_1, \\
f_2(v_n, v) + \varphi(v) - \varphi(v_n) + \frac{1}{r}\langle v - v_n, \mathcal{J}_2 v_n - \mathcal{J}_2 y_n \rangle \geq 0, \ \forall v \in \mathcal{E}_2, \\
x_{n+1} = \alpha_n x_n + (1 - \alpha_n)\mathcal{T}(u_n - \rho \mathcal{J}_1^{-1} g^* \mathcal{J}_3(g(u_n) - h(v_n))), \\
y_{n+1} = \alpha_n y_n + (1 - \alpha_n)\mathcal{S}(v_n + \rho \mathcal{J}_2^{-1} h^* \mathcal{J}_3(g(u_n) - h(v_n))), \ \forall n \geq 1,
\end{cases}
$$

*where $r \in (0, \infty)$, $(\|g\|^2 + \|h\|^2)^{-1} < \rho < 2(\|g\|^2 + \|h\|^2)^{-1}$ and $\{\alpha_n\}$ is a sequence in $[a, b]$ for some $a, b \in (0, 1)$.*
*If $\Gamma := \mathrm{SEFP}(\mathcal{T}, \mathcal{S}) \cap \mathrm{SEMEP}(f_1, f_2, \phi, \varphi) \neq \emptyset$, then we have the following:*

1. *$\{(x_n, y_n)\} \rightharpoonup (p, q) \in \Gamma$;*
2. *Furthermore, if $\mathcal{S}$ and $\mathcal{T}$ are semi-compact, then $\{(x_n, y_n)\} \to (p, q) \in \Gamma$.*

In Theorem 1, putting $\mathcal{B} = \mathcal{I}$, $\mathcal{E}_2 = \mathcal{E}_3$ and $\mathcal{J}_2 = \mathcal{J}_3$, then, by the similar proof in Theorem 1, the following result is obtained.

**Corollary 3.** *Let $\mathcal{E}_1$, $\mathcal{E}_2$, $f_1$, $f_2$, $\mathcal{T}$, $\mathcal{S}$ and $g$ be the same as above. Let $\{(x_n, y_n)\}$ be the iteration scheme in $E_1 \times E_2$ defined as follows: for any $(x_1, y_1) \in E_1 \times E_2$,*

$$
\begin{cases}
f_1(u_n, u) + \frac{1}{r}\langle u - u_n, \mathcal{J}_1 u_n - \mathcal{J}_1 x_n \rangle \geq 0, \ \forall u \in \mathcal{E}_1, \\
f_2(v_n, v) + \frac{1}{r}\langle v - v_n, \mathcal{J}_2 v_n - \mathcal{J}_2 y_n \rangle \geq 0, \ \forall v \in \mathcal{E}_2, \\
x_{n+1} = \alpha_n x_n + (1 - \alpha_n)\mathcal{T}(u_n - \rho \mathcal{J}_1^{-1} g^* \mathcal{J}_2(g(u_n) - v_n)), \\
y_{n+1} = \alpha_n y_n + (1 - \alpha_n)\mathcal{S}(v_n + \rho(g(u_n) - v_n)), \ \forall n \geq 1,
\end{cases}
$$

*where $r \in (0, \infty)$, $(\|g\|^2 + \|h\|^2)^{-1} < \rho < 2(\|g\|^2 + \|h\|^2)^{-1}$, $\{\alpha_n\}$ is a sequence in $[a, b]$ for some $a, b \in (0, 1)$.*

*If $\Gamma := \text{SEFP}(\mathcal{T}, \mathcal{S}) \bigcap \text{SEP}(f_1, f_2) \neq \varnothing$, then we have the following:*

1. *$\{(x_n, y_n)\} \rightharpoonup (p, q) \in \Gamma$;*
2. *Furthermore, if $\mathcal{S}$ and $\mathcal{T}$ are semi-compact, then $\{(x_n, y_n)\} \rightarrow (p, q) \in \Gamma$.*

## 4. Applications to the Split Equality Convex Minimization Problem

If $f = 0$ in the problem (MEP), then the mixed equilibrium problem reduces to the following *convex minimization problem* (shortly, (CMP)):

$$
\text{Find } x^* \in \mathcal{E}_1 \text{ such that } \phi(y) \geq \phi(x^*) \text{ for all } y \in \mathcal{E}_1.
$$

The solution set of the problem (CMP) is denoted by $\text{CMP}(\phi)$.

In the problem (SEMEP), if $f_1 = 0$, $f_2 = 0$, then the problem (SEMEP) reduces to the following *split equality convex minimization problem* (shortly, (SECMP)), which is formulated as follows: Find $x^* \in \mathcal{E}_1$ and $y^* \in \mathcal{E}_2$ such that

$$
\phi(x) \geq \phi(x^*), \ \varphi(y) \geq \varphi(y^*) \text{ for all } x \in \mathcal{E}_1, \ y \in \mathcal{E}_2 \text{ and } gx^* = hy^*.
$$

The solution set of the problem (SECMP) is denoted by $\text{SECMP}(\phi, \varphi)$, that is,

$$
\text{SECMP}(\phi, \varphi) = \{(x^*, y^*) \in \mathcal{E}_1 \times \mathcal{E}_2 : \phi(x) \geq \phi(x^*), \varphi(y) \geq \varphi(y^*), \ x \in \mathcal{E}_1, y \in \mathcal{E}_2, \ gx^* = hy^*\}.
$$

Therefore, Corollary 2 can be used to solve the problem (SECMP) and the following result can be directly deduced from Corollary 2.

**Theorem 2.** *Let $\mathcal{E}_1$, $\mathcal{E}_2$, $\mathcal{E}_3$, $f_1$, $f_2$, $\mathcal{T}$, $\mathcal{S}$, $g$ and $h$ be the same as above. Let $\phi$ and $\varphi$ be the same as in Definition 3. Let iteration scheme $\{(x_n, y_n)\}$ be defined as follows: for any $(x_1, y_1) \in \mathcal{E}_1 \times \mathcal{E}_2$,*

$$
\begin{cases}
\phi(u) - \phi(u_n) + \frac{1}{r}\langle u - u_n, \mathcal{J}_1 u_n - \mathcal{J}_1 x_n \rangle \geq 0, \ \forall u \in \mathcal{E}_1, \\
\varphi(v) - \varphi(v_n) + \frac{1}{r}\langle v - v_n, \mathcal{J}_2 v_n - \mathcal{J}_2 y_n \rangle \geq 0, \ \forall v \in \mathcal{E}_2, \\
x_{n+1} = \alpha_n x_n + (1 - \alpha_n)\mathcal{T}(u_n - \rho \mathcal{J}_1^{-1} g^* \mathcal{J}_3(g(u_n) - h(v_n))), \\
y_{n+1} = \alpha_n y_n + (1 - \alpha_n)\mathcal{S}(v_n + \rho \mathcal{J}_2^{-1} h^* \mathcal{J}_3(g(u_n) - h(v_n))), \ \forall n \geq 1,
\end{cases}
$$

*where $r \in (0, \infty)$, $(\|g\|^2 + \|h\|^2)^{-1} < \rho < 2(\|g\|^2 + \|h\|^2)^{-1}$ and $\{\alpha_n\}$ is a sequence in $[a, b]$ for some $a, b \in (0, 1)$.*

*If $\Gamma := \text{SEFP}(\mathcal{T}, \mathcal{S}) \bigcap \text{SECMP}(\phi, \varphi) \neq \varnothing$, then we have the following:*

1. *$\{(x_n, y_n)\} \rightharpoonup (p, q) \in \Gamma$;*
2. *Furthermore, if $\mathcal{S}$ and $\mathcal{T}$ are semi-compact, then $\{(x_n, y_n)\} \rightarrow (p, q) \in \Gamma$.*

**Remark 1.** *In Theorem 2, if we take $\mathcal{B} = \mathcal{I}$, $\mathcal{J}_2 = \mathcal{J}_3$ and $\mathcal{E}_2 = \mathcal{E}_3$, then, from Theorem 2, we can obtain some more convergence theorems to approximate a common element of the solution set of the split feasibility problem (SFP) and the solution set of the split convex minimization problem (SCMP).*

**Author Contributions:** Z.L.M., L.W. and Y.J.C. contributed equally in this work.

**Funding:** This research was funded by the National Natural Science Foundation of China Grant No. 11361070. This project was also supported by the Science Foundation of Education Department of Yunnan Province grant number 2018JS776.

**Acknowledgments:** The authors would like to thank the associate editor and the anonymous referee for his/her comments that helped us improve this article.

**Conflicts of Interest:** The authors declare no conflict of interest.

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
