# Peer review of "Some Results for Split Equality Equilibrium Problems in Banach Spaces"

_symmetry, doi:10.3390/sym11020194_

Reviewer 1 Report

see the enclosed report

Author Response

Response to Reviewer 1 Comments

Part A: Response to Reviewer 1

Point 1: page 2, line 25:  “then split equality problem”

Response 1: “then, the split equality” 

Point 2: page 2, lines 31-32: "SFP, SEP and split equality problems have   attracted the attention of many authors  in  Hilbert spaces. Some  iteration …"

Response 2: "The problems (SFP),  (SEQP)  and (SEP) in Hilbert spaces have attracted the attention of many  authors. Some iteration ...".

Point 3: page 3, lines 47-48: “... Banach space,  is nonexpansive, then the ...”

Response 3: “... Banach space  and  is nonexpansive, then the ...”

Point 4: page 3, Proposition 1: “A formal definition of smooth and uniformly smooth Banach space is in order”

Response 4: “Because it is common knowledge, so we don’t restate.”

Point 5: page 4, Lemma 4:“A formal definition of 2-uniformly smooth Banach

space is in order”

Response 5: “Because it is common knowledge, so we don’t restate.”

Point 6: page 5, relation (20): “It is not clear of why did the authors use the notation where in place of , where ” “A little explanation of this -if any-would be of interest.” 

Response 6: “ Because it is easy to use the famous Lemma 1 and Lemma 2, and the unified form with Lemma 1 and Lemma 2. 

Point 7: page 5, lines 100: “...it follows  that  is decreasing. So,  exists” 

Response 7: “... that the sequence  is decreasing; hence exists.”

                  But we don’t change  exists into exists in, because we unify the form in our article.

Point 8: pages 8-9, Relations (50)-(51): why we don’t use the stronger conclusion “ and ”.

Response 8: Because we just need the conclusions “ and ”

Part B: Modifed points by us

Point 1: . page 1, line 4: added the email address

Response 1: “University of Electronic Science and Technology of China, Chengdu, Sichuan, 611731, P.R. China”

Point 2: page 1, line 15: “The split feasibility problem(shortly, (SFP)) is to” was deleted

Response 2: page 1, line 15: “The split feasibility problem(shortly, (SFP)) is to” was deleted

Point 3: page 2, line 19: “...can be formulated as the SFP”

Response 3: “...can be formulated as the problem (SFP)…” 

Point 4: page 2, line 21: “…developed the SFP…”

Response 4: “…developed the problem (SFP)…” 

Point 5: page 2, line 21: “respectively,and let be real Hilbert space, …”

Response 5: “…respectively, be a real Hilbert space, …”

Point 6: page 2, line 22: “…that the split equality problem may reduce to the SFP…”

Response 6: “…that the problem (SEP) may reduce to the problem (SFP) …”

Point 7: page 2, line 26: “…The set of solutions of SEFP on T and S is…”

Response 7: “…The set of solutions of the problem (SEFP) on T and S is denoted as follows:”

Point 8: page 2, Definition 1: “…subsets of and respectively. Let , and be two bifunctions, andbe two bounded linear operators…”

Response 8: “…subsets of ,,respectively. Let ,,betwo bifunctions and,be two bounded linear operators…”

Point 9: page 2, line 31: “…The research on SFP, split equality problem, split common null point…”

Response 9: “…The recent research on the probelms (SFP), (SEP), the split common null point…”

Point 10: page 3, line 37-38: “…we can not find out the results on the split 37 equlibrium problem and split equality equilibrium problem in Banach spaces” 

Response 10: “…we can not find out the results on the problems (SEP) and (SEQP) in Banach spaces.”

Point 11: page 3, line 40-41: “…find a common element of split equality fixed point problem for nonexpansive mappings and split equality equilibrium problem in three Banach spaces…”

Response 11: “…find a common element of the problem (SEFP)  and the  problem (SEEP)for nonexpansive mappings in three Banach spaces.”

Point 12: page 3, line 49-51: “…By the Hahn-Banach theorem,is nonempty (see [28] for more details). If…”

Response 12: “…Note that, by the Hahn-Banach theorem,  is nonempty (see [28] formore details). If…”

Point 13: page 3, line 54 and 56:” (1) and (2)”

Response 13: “1 and 2”

Point 14: page 5, line 93: “…then”

Response 14: “…then we have the following:”

Point 15: page 5, line 99: “We divide the proof of conclusion 1 into four steps”

Response 15: “1. For 1, we divide the proof of the conclusion 1 into four steps as follows:”

Point 16: page 5, line 100: “Taking, from Lemma 2, we know that  and . Furthermore,”

Response 16: “In fact, Taking, from Lemma 2, we know that  and. Furthermore, we have”

Point 17: page 6, line 100: “ …using Lemmas 3 and Lemma 4…”

Response 17: …using (2), (3), Lemma 3 and Lemma 4, we have”

Point 18: page 6, line 100: “… it followsfrom the fourth equality in (21) that”

Response 18: “…it follows from (1) that” 

Point 19: page 6, line 100: “…we know that  and so…” 

Response 19: “…we know that  and so, by (4) and (5),” 

Point 20: page 6, line 103(step 2): “…It follows from (27) that” 

Response 20: “…In fact, it follows from (7) that” 

Point 21: page 7 , in (28): “…for some,we have” 

Response 21: “…for some,by (8), we have” 

Point 22: page 7, in (30): “Applying the properties ofand Lemma 3, we have” 

Response 22: “Applying the properties of,(9)and Lemma 3, we have” 

Point 23: page 7, in (41): “we know that” 

Response 23: “by (12) and (13), it follows that” 

Point 24: page 8, in (45): “So from (44) and (45), we obtain” 

Response 24: “ So,it follows from (15) and (16) that” 

Point 25: page 8, in (47): “It follows from (21) and lemma 4 that” 

Response 25: “Also, it follows from (1) and Lemma 4 that” 

Point 26: page 8, in (47): “Adding inequality (48) and inequality (49) and taking into account the fact that, we obtain that” 

Response 26: “Adding the inequalities (18), (19) and taking into account the fact  that, we obtain” 

Point 27: page 9, in (55): “By (30), (42), (52) and (55), we have” 

Response 27: “By (14), (21) and (22), we have” 

Point 28: page 9, in (58): “it follows from (52), (56) and (58) that” 

Response 28: “it follows from (21) and (23) that” 

Point 29: page 9, in (60): “and so it follows from (53), (57) and (60) that” 

Response 29: “and so it follows from (21) and (24) that” 

Point 30: page 10, line 105-107: “Since  and are reflexive,…so we may assume that…are demiclosed and so, from Lemma…we know that…” 

Response 30: “ In fact, since and are reflexive,…we may assume that…aredemiclosed, so, from Lemma…it follows that…” 

Point 31: page 10, in (64): “Applying (C1) and (C4), we get that” 

Response 31: “Applying the conditions (C1) and (C4), it follows that” 

Point 32: page 10, in (65): “…from(C3), it yields that” 

Response 32: “from the condition (C3), it follows that”

Point 33: page 11, line 115: “Since S and T are semi-compact…” 

Response 33: “In fact, since S and T are semi-compact…” 

Point 34: page 11, line 121: “…So, the equilibrium problem (1) reduces to the problem:” 

Response 34: “…So, the problem (EP) reduces to the problem:”

Point 35: page 11, line 123: “…The set of solutions of the problem (71) is denoted…” 

Response 35: “…The set of solutions of the problem (GMEP) is denoted…” 

Point 36: page 11, line 124: “…If in (70), then (70) reduces to the problem…The set of solutions of the problem (72) is denoted…” 

Response 36: “…Ifin the problem (GMEP),  then the problem (GMEP)  reduces to theproblem…The set of solutions of the problem (MEP) is denoted…”

Point 37: page 12, line 130: “…then” Response 37: “…then we have the following:”

Point 38: page 12, line 132: “…then the split equality generalized mixed equilibrium problem (73) reduces to the following so called the split equality mixed…The set of solutions of the problem (76) is denoted…” 

Response 38: “…then the problem (SEGEMP) reduces to the following so called the splitequality mixed…The set of solutions of the problem (SEMEP) is denoted…” 

Point 39: page 12, line 136: “…then” 

Response 39: “…then we have the following:” 

Point 40: page 13, line 143: “…then” 

Response 40: “…then we have the following:” 

Point 41: page 13, line 147: “If in (72), then… convex minimization problem…The solution set of the problem (80) is denoted…” 

Response 41: “…If in the problem (GMEP), then …convex minimization problem (shortly, (CMP))…The solution set of the problem (CMP) is denoted…” 

Point 42: page 13, line 148: “In (76), if…then the split equality mixed equilibrium problem (76) reduces to…minimization problem…The solution set of the problem (81) is denoted…solve the split equality convex minimization problem (81) …” 

Response 42: “In the problem (SEMEP), if…then the  problem (SEMEP) reduces to…minimization problem (shortly, (SECMP))…The solution set of the  problem (SECMP) isdenoted…solve the  problem (SECMP)…” 

Point 43: page 13, line 148: “…then” 

Response 43: “…then we have the following:” 

Point 44: page 13, line 156-157: “…the split feasibility problem (3) and the solution set of the split convex minimization problem”

Response 44: “…the split feasibility problem (SFP) and the solution set of the  split convex minimization problem (SCMP).” 

Reviewer 2 Report

 Comments on the paper:
Some Results on Split Equality Equilibrium Problems in Banach Spaces
by Zhaoli Ma, Lin Wang and Yeol Je Cho
The paper under review is a sequel to [24] (Z. L. Ma, L. Wang, S. S. Chang, W. Duan Fixed Point Theory Appl. 2015, 2015:31), where similar results concerning split equality equilibrium problems were proved in Hilbert spaces. In the present paper, two Hilbert spaces are replaced by uniformly convex and 2-uniformly smooth Banach spaces satisfying Opial's property, and the third one is
replaced by a smooth, reflexive and strictly convex Banach space. This requires some modifications in the presentation of the results and arguments. I think this paper merits publication but I am not sure if the journal "Symmetry" is appropriately chosen.
Minor correction:
page 5, line 98: should read "the iteration scheme (20) is well defined".

Author Response

Point 1: page 5, line 98: "the iteration scheme (21) is well defined"

 Response 1: the iteration scheme (1) is well defined”

2.  The numbers of the equalities and inequalities have been removed in our article

This manuscript is a resubmission of an earlier submission. The following is a list of the peer review reports and author responses from that submission.